# Distal Renal Tubular Acidosis in Patients with Autoimmune Diseases—An Update on Pathogenesis, Clinical Presentation and Therapeutic Strategies

**DOI:** 10.3390/biomedicines10092131

**Published:** 2022-08-31

**Authors:** Oana Ungureanu, Gener Ismail

**Affiliations:** 1Department of Nephrology, Fundeni Clinical Institute, 022238 Bucharest, Romania; 2Department of Nephrology, Carol Davila University of Medicine and Pharmacy, 020021 Bucharest, Romania

**Keywords:** renal tubular acidosis, systemic lupus erythematosus, Sjögren syndrome

## Abstract

Distal renal tubular acidosis (DRTA) has been reported in association with autoimmune diseases, such as Sjögren’s syndrome, systemic lupus erythematosus (SLE), autoimmune hepatitis, primary biliary cirrhosis, rheumatoid arthritis and autoimmune thyroiditis. Whether we talk about the complete or incomplete form of DRTA associated with autoimmune diseases, the real incidence is unknown because asymptomatic patients usually are not identified, and most of the reported cases are diagnosed due to severe symptoms secondary to hypokalemia, a frequent finding in these cases. The mechanisms involved in DRTA in patients with autoimmune diseases are far from being fully elucidated and most of the data has come from patients with Sjögren’s syndrome. This review will present different hypotheses raised to explain this association. Also, aiming for a better understanding of the association between autoimmune diseases and DRTA, our review summarizes data from 37 case reports published in the last five years. We will emphasize data regarding clinical presentation, biological alterations, treatment and outcome. A very important question is whether immunosuppressive therapy is helpful in DRTA associated with autoimmune diseases. Because the pathology is rather rare, treatment is not standardized, and reported results are often contradictory. Corticosteroids are frequently used, but multiple other immunosuppressive drugs have been proposed and will be approached in this review.

## 1. Introduction

Distal renal tubular acidosis (DRTA) is characterized by hyperchloremic metabolic acidosis with normal anion gap and inability to acidify the urine to a pH lower than 5.3. Distal renal tubular acidosis can evolve without symptoms and systemic acidosis, with this form being defined as incomplete DRTA. The incomplete form necessitates the use of a urinary acidification test for establishing the diagnosis. The prevalence of the DRTA in patients with autoimmune diseases is unknown, especially if we refer to the incomplete form. We will present a synthesis of 37 case reports published recently in the literature with this association, emphasizing the most important clinical and biological modification. We will also discuss data regarding possible pathogenic mechanisms, treatment and outcome.

## 2. Epidemiology, Clinical Presentation and Diagnosis

The prevalence of DRTA in patients with Sjögren’s syndrome is quite variable depending on different publications ranging between 6.8% and 70% [1,2,3,4]. The association of SLE and DRTA was recently published in a systematic review (2020) which identified only 26 cases reported in literature [5]. Two additional case reports with this association were published in 2020, and another study published recently showed a prevalence of 14.81% of DRTA (16 patients) in a cohort of 108 patients with SLE [6,7,8]. Generally, the majority of cases with complete DRTA published in literature are diagnosed due to severe symptoms associated with hypokalemia, such as quadriparesis or even respiratory arrest, and therefore, the real prevalence of DRTA associated with autoimmune diseases is difficult to estimate [6,9,10,11,12,13,14,15,16,17,18,19,20,21,22]. Therefore, urinary acidification tests are essential in diagnosing those patients who did not develop any symptoms and would otherwise escape undiagnosed. The gold standard is the ammonium chloride test, but it has important digestive side effects. An alternative acidification test is the furosemide/fludrocortisone test, which is better tolerated, but less sensitive and accurate than the ammonium chloride test, so it would be better used as a screening tool, given its better tolerability and its reasonable negative predictive value [23].

Aiming for a better understanding of the disease, we performed a search of the PubMed database using MeSH descriptors (Acidosis, Renal Tubular; Lupus Erythematosus, Systemic; Sjogren’s Syndrome, Systemic Vasculitis, Rheumatoid Vasculitis, Arthritis, Rheumatoid, Anti-Neutrophil Cytoplasmic Antibody-Associated Vasculitis, Churg-Strauss Syndrome, IgA Vasculitis, Spondylitis, Ankylosing, Cryoglobulins, Hepatitis, Autoimmune, Liver Cirrhosis, Biliary), and we identified 37 individual case reports published since December 2016 with the association of distal renal tubular acidosis and the following autoimmune disorders: SLE, Sjögren’s syndrome, autoimmune hepatitis, primary biliary cirrhosis and rheumatoid arthritis (see Table 1) [6,10,11,13,15,16,17,18,20,24,25,26,27,28,29,30,31,32,33,34,35,36,37,38]. We also identified a case of hereditary autoimmune polyendocrine syndrome (DRTA, hypoparathyroidism, Addison disease), published in a series of cases, which was not included in our analysis, due to the lack of individual data availability [39]. It is important also to mention that we have not found any case report with the association of pANCA or cryoglobulinemic vasculitis with DRTA. In the majority of cases, the diagnosis was of Sjögren’s syndrome, with 4 cases out of 37 being diagnosed with SLE or overlap syndrome (SLE and Sjögren’s syndrome) and only 4 cases out of 37 being diagnosed with another autoimmune disease (2 patients with primary biliary cirrhosis, 1 patient with autoimmune hepatitis and 1 patient with rheumatoid arthritis). We also included one patient with the association of Sjögren’s syndrome and Hashimoto thyroiditis and celiac disease. Most of the patients were females (35 out of 37), with age ranging from 16 years old to 67 years old, most of them being in the third or fourth decade of life (mean age 33.57 ± 13.25 years old). The common point of all these reported cases was the severe hypokalemia (mean 2.27 ± 0.60 mmol/L, ranging from 1.40 mmol/L to 3.40 mmol/L). Consequently, most of the patients (26 out of 37) presented with symptoms associated with hypokalemia (progressive muscular weakness and eventually hypokalemic periodic paralysis). In a limited number of cases (3 out of 37), hypokalemic paralysis led to respiratory failure, eventually with orotracheal intubation and ventilation [20,24,25]. In one particular case, respiratory failure with the need of orotracheal intubation occurred after producing brisk diuresis and consequent severe hypokalemia following nephrostomy for hydronephrosis caused by ureteral lithiasis [24]. Although symptoms related to hypokalemia were the most frequently reported, these patients had multiple other manifestations. Since most of the patients were diagnosed with Sjögren’s syndrome, many of them presented with SICCA symptoms (xerostomia, xerophthalmia). Also, manifestations related to calcium metabolism, such as hypocalcemia, osteomalacia, renal/ureteral lithiasis, nephrocalcinosis or fractures were also frequently reported (20 patients out of 37 had one or more of these manifestations; see Table 1). A smaller proportion of patients manifested hematological alterations (anemia–6 patients, thrombocytopenia–3 patients, purpura–2 patients). Since most of the patients are diagnosed at fertile age, it is also important to mention the obstetrical complications, although these are not frequently reported. In this series of 31 case reports, we identified two patients who had a history of miscarriages or stillbirth, one of them being diagnosed with overlap syndrome (SLE and Sjögren’s syndrome) [16,24]. Another patient was diagnosed with DRTA during pregnancy, which evolved with multiple complications, respectively preterm labor (at 36 weeks of gestation), fetal bradyarrhythmia and fetal growth retardation [26].

Regarding renal involvement, patients presented with a mean eGFR of 73.29 ± 31.98 mL/min/1.73 m^2^, ranging from 23 mL/min/1.73 m^2^ to 126 mL/min/1.73 m^2^. All the patients had persistent alkaline urinary pH, with a minimum value of 6 and a mean value of 7.06 ± 0.567. Most of the patients were diagnosed with the complete form of DRTA (with systemic acidosis), although 4 out of 33 patients had serum HCO_3_^−^ values of a minimum of 20 mmol/L, approaching the normal range. One of them actually had the incomplete form of DRTA (pH 7.35, HCO_3_^−^ 22 mmol/L). Even though they had minimal or no systemic acidosis, these patients associated severe hypokalemia (values as low as 1.5 mmol/L), with hypokalemic periodic paralysis. Kidney biopsy was available only for nine patients, showing, as expected, variable degrees of tubulo-interstitial nephritis.

## 3. Pathogenesis

The mechanisms involved in impairing distal tubular acidification in patients with autoimmune diseases are far from being fully elucidated, and most of the data are from patients with Sjögren’s syndrome. Since 1992, there have been a few reports that raised different hypotheses to explain this association. The mechanisms seem to imply the absence of the H^+^-ATPase in the connecting tubules and collecting ducts of the kidney in patients with Sjögren’s disease and DRTA [40,41,42,43,44,45]. This was proven using anti-H^+^-ATPase antibodies in immunofluorescence studies, which show no reactivity in patients with Sjögren’s syndrome and DRTA compared to control patients where reactivity was seen. More questions were raised starting from this observation, as described below (see Figure 1).

A.Is the paucity or deficiency of the H^+^-ATPase expression directly correlated to histological damage in patients with autoimmune diseases? Ha Yeon Kim et al., and De Franco et al., reported that a diffuse interstitial inflammatory infiltrate composed by lymphocytes and plasma cells was seen in light microscopy in the renal tissue of two patients with Sjögren’s syndrome and DRTA, one of which also showed lesions of tubulitis [40,41]. When electron microscopy was performed, the ultrastructural features supported the presence of intact intercalated cells [40,42,44]. The absence of the H^+^-ATPase in immunofluorescence with diffuse interstitial inflammatory infiltrate under light microscopy and the presence of intact intercalated cells on electron microscopy images suggest that the acidification defect may be due the functional lesions of intercalated cells and polarity changes, possibly determined by inflammatory processes, with the lack of expression of H^+^-ATPase.B.The second question is whether or not these patients with autoimmune diseases and DRTA have other defects expressing channels/ion pumps/enzymes implicated in urinary acidification (like the anion exchanger 1 (AE1)/band 3 protein, pendrin or type II carbonic anhydrase). Both Walsh et al., and De Franco et al., proved that renal tissue from two patients with Sjögren’s syndrome and DRTA showed complete lack of immunoreactivity for basolateral AE1 and apical H^+^-ATPase in intercalated cells, although there was reactivity for AE1 in blood cells captured in the tissue sample derived from these patients [41,45]. Ha Yeon Kim et al., also showed that kidneys of patients with Sjögren’s syndrome revealed a weak fluorescence for pendrin compared to normal kidney tissue, which showed a bright distinct red fluorescence reaction [40]. So, these patients with Sjögren’s syndrome and DRTA seem to manifest multiple defects in the expression of protein channels or pumps, which are involved in distal urinary acidification.C.The third question derived from these observations is that the lack of expression of structures or molecules like H^+^-ATPase/AE1/pendrin/type II carbonic anhydrase is due to autoantibodies directed against the intercalated cells or directly to these transporters or enzymes involved in distal urinary acidification. Devuyst et al., demonstrated that G immunoglobulins found in the serum of one patient diagnosed with Sjögren’s syndrome and DRTA reacted with intercalated cells in collecting duct profiles of human control kidneys [42]. In return, De Franco et al., and Cohen et al., incubated serum from two patients with Sjögren’s syndrome and DRTA with control kidney tissue samples, and no reactivity was seen [41,44]. Cohen et al., also incubated patient’s serum with his own kidney tissue sample, with no reactivity being seen, in contrast with De Franco et al., who did not perform this procedure; therefore, the authors highlight the fact that even though there was no reactivity of patient’s serum with control kidney sample, a possible reaction with patient’s renal tissue cannot be excluded [41,44]. Also, the authors raise the hypothesis that inflammatory damage to intercalated cells might lead to exposure of intracellular antigens, which stimulates autoantibodies production, which may be just a marker of intercalated cell damage and not actually responsible for the injury to these cells. Chao et al., also demonstrated the presence of antibodies to renal tubular epithelial cells and antibodies to the B1/B2 subunit of H^+^-ATPase in the serum of 6 out of 11 patients diagnosed with DRTA (ten of them associated Sjögren’s syndrome) [46]. Patients without DRTA, including Sjögren’s syndrome patients and normal control patients, were negative for these autoantibodies [46]. In return, although the patient’s kidney sample in the case reported by Cohen et al., lacked the expression of H^+^-ATPase, the presence of autoantibodies to H^+^-ATPase subunits or to intercalated cells was not proved when patient’s serum was incubated with purified bovine kidney vacuolar H^+^-ATP-ase or with normal human kidney tissue [44]. Interesting data regarding a possible role of autoantibodies in reducing the expression of structures involved in distal renal acidification came from a recent published case with the association of primary biliary cirrhosis and DRTA [35]. First, the authors proved the lack of expression of AE1, B1 and a4 subunits of H^+^-ATP-ase by immunofluorescence staining of kidney biopsy sections [35]. After that, they incubated kidney sections from healthy donors with either serum from the patient or serum from healthy donors [35]. Reactivity of patient serum with the otherwise healthy tissue was proven using anti-human IgG fluorescent labelled antibodies, which showed specific labeling of the apical pole of AQP2 and AE1 positive collecting duct cells [35]. It is important to mention that this patient associated an active form of tubulointerstitial nephritis [35]. However, the authors state that it is difficult to appreciate whether these autoantibodies directed at collecting duct cells are the cause or consequence of dRTA and tubulointerstitial nephritis, and further studies are necessary to clarify this.

As far as other proteins involved in distal urinary acidification, Takemoto et al., proved that patients with Sjögren’s syndrome can also develop autoantibodies directed to type II carbonic anhydrase [47]. The authors proposed a logistic regression model, which showed the association between anti-type II carbonic anhydrase antibody levels and the presence of DRTA (odds ratio (OR) 1.9 per 0.1 ELISA unit/mL increase; 95% confidence interval (CI): 1.2 to 2.8 per 0.1 ELISA unit/mL increase; *p* = 0.02) [47]. Anti-type II carbonic anhydrase antibody levels also correlated with urinary β2-microlobulin levels, which is considered a marker of tubular and interstitial injury and the authors raised the hypothesis that inflammatory damage to tubular cells might lead to exposure of intracellular antigens like type II carbonic anhydrase and secondary production of autoantibodies [47]. Therefore, the presence of type II carbonic anhydrase autoantibodies would be just a marker of tubular damage. However, there are in vitro studies which point to the fact that type II carbonic anhydrase autoantibodies might actually inhibit the catalytic activity of the type II carbonic anhydrase and cause DRTA, especially in the inflammatory renal environment of Sjögren’s syndrome patients, which might allow these autoantibodies to reach the cytosolic compartment of the cells [48].

Another interesting discussion regarding the possibility that autoantibodies react and diminish the expression and action of channels, pumps or enzymes which are involved in distal renal tubular acidification is if the inflammatory infiltrate seen in kidney biopsy tissue in many cases of Sjögren syndrome, for example, might influence the autoantibody production. Th17 cells, a subset of CD4+ lymphocytes, were identified for the first time in murine models of autoimmunity, being named after their signature cytokine IL-17 [49]. After that, Th17 cells and IL-17 have been related to the pathogenesis of multiple autoimmune diseases, such as rheumatoid arthritis, SLE and Crohn’s disease [49]. It appears that Th17 and IL-17 mediate the inflammatory response, favoring inflammatory infiltration in the targeted tissue and also that this subtype of lymphocytes promotes autoantibodies production [49]. Interesting observations came from patients with SLE, where IL-17 producing cells were discovered in kidney biopsies from patients with lupus nephritis, and also Th17 cells were related to activation of B lymphocytes, causing an increased autoantibodies production [49]. It would be interesting to research if Th17 cells are found in renal biopsy tissue derived from patients with the association of DRTA and autoimmune diseases and if they might be related to production of autoantibodies directed at structures involved in distal urinary acidification.

Also, another possible direction for research would be the IL-31/IL-33 axis, which may be involved in autoimmunity, as recent studies suggest [50]. For example, in SLE, IL-33 may play an important role in the acute phase of the disease [50]. In the case with primary biliary cirrhosis, kidney biopsy tissue analysis showed acute tubulointerstitial nephritis [35]. It would be interesting to research if IL-33 may have a role in DRTA development and evolution in this type of case, with important acute lesions on kidney biopsy tissue.

## 4. Treatment and Outcome

A very important question is whether immunosuppressive therapy is helpful in DRTA. It should have a role in slowing the evolution of the disease by at least two mechanisms: by reducing the interstitial inflammatory infiltrate that characterizes tubulo-interstitial nephritis associated with Sjögren’s syndrome and by slowing autoantibodies production. Nevertheless, because the pathology is rather rare, treatment is not standardized and reported results are often contradictory. Corticosteroids are the most frequently used therapy, but multiple other immunosuppressive or immunomodulatory drugs have been proposed with or without corticosteroids, including hydroxychloroquine, rituximab, cyclophosphamide, methotrexate, mycophenolate mofetil, azathioprine [1,2,51,52]. Oftentimes, additional immunosuppressive therapies other than corticosteroids are used in those severe cases that might present with concomitant glomerular and tubular involvement in Sjögren’s syndrome (membranoproliferative glomerulonephritis secondary to cryoglobulinemia). In most reports, immunosuppressive therapy improves kidney function in patients with Sjögren’s syndrome and tubulointerstitial nephritis. Jasiek et al., compared the effect on eGFR of corticosteroids associated with other immunosuppressive drugs versus corticosteroid monotherapy and also the effect of rituximab versus corticosteroids alone in a cohort of patients with Sjögrens’s syndrome and tubulo-interstitial nephritis and no significant difference was reported if an eGFR gain of 20 mL/min/1.73 m^2^ was taken into account as outcome [53]. The overall eGFR value increased after six months of follow-up from 39.8 to 49.3 mL/min/1.73 m^2^ (*p* < 0.001) [53]. In a study proposed by Shen et al., comparing the effect of corticosteroids combined with cyclophosphamide versus corticosteroids alone in patients with Sjögrens’s syndrome and tubulo-interstitial nephritis, eGFR rose with 21.35 ± 19.63 mL/min/1.73 m^2^ vs. 2.72 ± 19.11 mL/min/1.73 m^2^ after 12 months of follow-up (*p* < 0.001) in the group receiving cyclophosphamide [54]. Therefore, it is not clear whether corticosteroids are the best option for slowing down the inflammation seen in tubulointerstitial nephritis associated with Sjögren’s syndrome or if other immunosuppressive drugs bring benefit in these cases. Regarding immunosuppressive therapy for DRTA, data are even more dispersed with corticosteroids being the most frequently used (see Table 2).

In the 37 individual case reports published in the last 5 years, patients received potassium and bicarbonate supplements, with or without additional immunosuppressive therapy. Corticosteroids (usually oral prednisone) were indeed the most frequently used immunosuppressive drugs, being administered in 17 cases out of 33, for whom data regarding treatment was available. Hydroxychloroquine was also frequently used (13 cases out of 33). The effect was variable. For most of the patients, acidosis and hypokalemia improved after potassium and bicarbonate supplementation and immunosuppressive drug administration, but for some of the patients, it is mentioned that they remained dependent on oral potassium and bicarbonate supplements. This suggests that although immunosuppressive therapy has an important effect on preserving or ameliorating renal function and interstitial infiltrate, it might have no effect on treating the hypokalemia and acidosis. Three patients received a combination of azathioprine, prednisone and hydroxychloroquine [18,24,32]. In all of these cases, it is mentioned that acidosis and hypokalemia were not ameliorated and the patients remained dependent on oral supplementation with potassium and bicarbonate. However, there are also some examples in which immunosuppressive therapy seems to improve distal tubular function because, after a period of treatment, oral supplementation for potassium and alkali was no longer needed [11,28,32]. One of the patients received only hydroxychloroquine, another patient received a combination of hydroxychloroquine and prednisone and the last patient received rituximab. This last case report is interesting to mention because initially the patient was treated with azathioprine in combination with prednisone and hydroxychloroquine, but no amelioration of DRTA was observed. Therefore, azathioprine was replaced by rituximab (1 g–day 0 and day 14; followed by courses administered every 6 months). After 4 years of treatment, eGFR improved from 46 mL/min/1.73 m^2^ to 71 mL/min/1.73 m^2^ and the patient no longer needed oral potassium and alkali supplements suggesting that DRTA might respond to Rituximab. Another immunosuppressive drug used for two patients was mycophenolate-mofetil, without signs of DRTA improvement [17,33]. In only one case out of these 33 reports, cyclophosphamide was used, with eGFR improvement and potassium value amelioration [33]. Immunosuppressive therapy was not used in those three cases of autoimmune hepatitis, primary biliary cirrhosis and rheumatoid arthritis included in our analysis [36,37]. A meta-analysis published in 2021 which included patients with renal tubular acidosis and liver pathology (autoimmune hepatitis and alcoholic hepatitis) showed that corticosteroids improved short term mortality in this group, but it is not mentioned any effect on the evolution of DRTA [55]. Therefore, data regarding immunosuppressive therapy is still scarce, with variable results.

Regarding new possibilities of treatment, blocking IL-17 activity with the new monoclonal antibody secukinumab might be an alternative for patients with SLE or Sjögren’s syndrome [56]. The molecule has been already adopted for autoimmune diseases like psoriatic arthritis and ankylosing spondylitis and the important role that Th17 cells and IL-17 have in maintaining the inflammatory response and antibodies production was proven in SLE and it has been detailed above. So, it would be interesting to research if targeting IL-17 in patients with autoimmune diseases and DRTA has any effect on the manifestations related to DRTA.

## 5. Conclusions

Distal renal tubular acidosis is one of the important comorbidities in patients with autoimmune diseases, in some cases having severe or life-threatening presentation secondary to ionic disturbances or acidosis. Distal renal tubular acidosis is described mostly in patients with Sjögren’s syndrome, but might be diagnosed in patients with other autoimmune diseases, including SLE. Complex mechanisms are involved in the pathogenesis of the DRTA in autoimmune diseases and the treatment of these patients is challenging, including potassium management, corticosteroids, immunosuppressants, rituximab with variable results and further research is needed for a comprehensive understanding of the pathology and for identifying the most efficient treatment options.

## Figures and Tables

**Figure 1 biomedicines-10-02131-f001:**
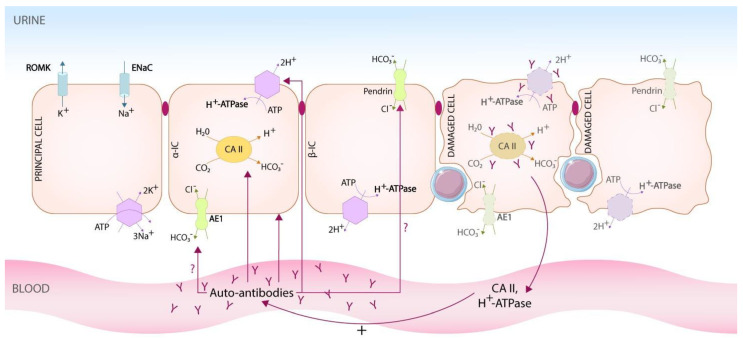
**Pathogenesis of DRTA in autoimmune diseases.** Most of the data published until now came from studies on patients with Sjögren’s syndrome. It has been proved that in patients with DRTA, expression of H^+^-ATPase, pendrin and AE1 by intercalated cells is altered/non-existent, even though the ultrastructural aspect of these cells is not modified when it is explored by electron microscopy. The lack of expression of different transporters implied in urinary acidification seems to be a result of a polarity loss in intercalated cells, rather than a decrease in their number. The presence of autoantibodies to intercalated cells, H^+^-ATPase and CA II has been proven in patients with DRTA and Sjögren’s syndrome. It is not clear if these autoantibodies are just a consequence of cellular damage with releasing of antigens in blood (subunits of CA II, H^+^-ATPase) and immune stimulation with antibodies production or if the autoantibodies have pathogenic effects and affect the activity of the targeted molecules. In case of CA II, the pathogenic effect has been proven. ɑ-IC, alpha intercalated cell; β-IC, β intercalated cell; AE1, anion exchanger 1; CA II, carbonic anhydrase II.

**Table 1 biomedicines-10-02131-t001:** Summary of recently published case reports, which describe the association of dRTA and autoimmune diseases.

Nr.	Author–Year of Publication	Gender	Age [Years]	Diagnosis	Presentation	Blood pH	Serum HCO_3_^−^ [mmol/L]	Serum K^+^ [mmol/L]	Creatinine [mg/dL]; eGFR [mL/min/1.73 m^2^]	Urine pH	Kidney Biopsy	Treatment	Outcome
1	Abdulla et al.–2017 [27]	F	57	Sjögren syndrome	SICCA, Fracture, hypocalcemia, anemia (9.8 g/dL)	NS	13	2.8	1.1; 56	6.5	-	K^+^, HCO_3_^−^ supplements; HCQ	K^+^ correction
2	Ammad Ud Din et al.–2020 [6]	F	44	SLE	Hypokalemic paralysis, SICCA, proteinuria	7.21	10	1.6	0.8; 90	6.5	TIN	K^+^, HCO_3_^−^ supplements; Prednisone 20 → 40 mg/day, HCQ	K^+^ correction; Proteinuria remission
3	Basok et al.–2021 [24]	F	32	Sjögren syndrome + SLE	Hypokalemic paralysis with respiratory failure, SICCA, hypocalcemia, lithiasis, nephrocalcinosis, miscarriages	7.26	8.1	1.6	1.04; 71	6.5	-	K^+^, HCO_3_^−^ supplements; HCQ, prednisone, AZA	No improvement of acidosis
4	Berrhoute et al.–2019 [15]	F	30	Sjögren syndrome	Hypokalemic paralysis, hypocalcemia, proteinuria (1.57 g/24 h)	NS	20	1.5	1.43; 49	6.5	TIN	K^+^, HCO_3_^−^ supplements; HCQ, CS	NS
5	Jackson et al.–2021 [11]	F	38	Sjögren syndrome	Hypokalemic paralysis, SICCA, hyperthyroidism (Graves disease)	7.25	NS	1.6	0.92; 79	8	-	K^+^, HCO_3_^−^ supplements; HCQ	Acidosis and K^+^ correction (no need for supplements)
6	Jung et al.–2017 [28]	F	60	Sjögren syndrome	Fracture, associated Fanconi syndrome	7.3	17.2	2.7	1.6; 35	7	TIN	K^+^, HCO_3_^−^ supplements; HCQ	Acidosis and K^+^ correction; creatinine increase
7	Jung et al.–2017 [28]	F	19	Sjögren syndrome	Muscular weakness, hypocalcemia	7.37	7.5	1.7	0.7; 126	7.5	-	K^+^, HCO_3_^−^ supplements; HCQ, CS	Acidosis and K^+^ correction (no need for supplements)
8	Ho et al.–2019 [17]	F	44	Sjögren syndrome	SICCA	7.29	16	2.8	1.31; 50	6.5	Severe TIN	K^+^, HCO_3_^−^ supplements; MMF 750 mg × 3/day, prednisone 1 mg/kg	Initial increase in creatinine (1.95 mg/dl), with subsequent decrease after CS and MMF initiation; long term need for supplements.
9	Louis-Jean et al.–2020 [13]	F	19	Sjögren syndrome	Hypokalemic paralysis, SICCA, proteinuria	7.21	17	2.1	NS	7	-	K^+^, HCO_3_^−^ supplements;	Repeated episodes of hypokalemia and paralysis
10	Martinez-Granados et al.–2017 [25]	F	38	Sjögren syndrome	Hypokalemic paralysis, respiratory failure, nephrocalcinosis	7.12	12	1.9	In normal range	8	-	K^+^, HCO_3_^−^ supplements; HCQ	NS
11	Mbengue et al.–2021 [10]	F	20	Sjögren syndrome	Hypokalemic paralysis (tetraparesis), nephrocalcinosis	NS	15.69	1.4	0.7; 125	7.5	-	K^+^, HCO_3_^−^ supplements; HCQ, CS	Acidosis and K^+^ correction
12	Monteiro Queiroz et al.–2020 [29]	F	24	Sjögren syndrome	Hypokalemic paralysis, nephrocalcinosis, lithiasis with hydronephrosis, purpura, SICCA	7.16	9.8	1.4	1.34; 55	7.5	-	K^+^ citrate; prednisone 40 mg/day	K^+^ correction
13	Narayan et al.–2018 [30]	F	25	Sjögren syndrome	Bleeding diathesis, severe thrombocytopenia (10,000/µL), severe anemia (6.8 g/dL)	NS	NS (nAGMA)	2.4	1; 78	7.5	-	K^+^ supplements; HCQ, CS	Thrombocytopenia and K^+^ correction
14	Narayan et al.–2018 [30]	F	28	Sjögren syndrome	Thrombocytopenia, anemia	NS	NS (nAGMA)	2.6	In normal range	7	-	K^+^ supplements; HCQ, CS	Thrombocytopenia and K^+^ correction
15	Narayan et al.–2018 [30]	F	30	Sjögren syndrome	Hypokalemic paralysis (tetraparesis), SICCA	NS	NS (nAGMA)	1.9	In normal range	6.5	-	K^+^, HCO_3_^−^ supplements; prednisone	Muscular weakness improvement
16	Paliwal et al.–2018 [31]	M	29	Sjögren syndrome + SLE	Hypokalemic paralysis, SICCA, malar rash, proteinuria (2.1 g/24 h), lithiasis, fever	7.28	8.3	1.8	NS	7.5	TIN	K^+^, HCO_3_^−^ and Ca^2+^ supplements; prednisone	Improvement of proteinuria
17	Paliwal et al.–2018 [31]	F	16	Sjögren syndrome	Hypokalemic paralysis, nephrocalcinosis, lithiasis, SICCA	7.26	12	2.1	NS	7	-	K^+^, HCO_3_^−^ and Ca^2+^ supplements;	K^+^ correction
18	Paliwal et al.–2018 [31]	M	25	Sjögren syndrome	Hypokalemic paralysis, SICCA	7.35	22	2.53	NS	6.5	-	K^+^, HCO_3_^−^ and Ca^2+^ supplements;	Acidosis and K^+^ value improvement
19	Paliwal et al.–2018 [31]	F	27	Sjögren syndrome	Hypokalemic paralysis, SICCA, proteinuria (0.64 g/24 h)	7.24	10.2	2.7	NS	6	-	K^+^, HCO_3_^−^ and Ca^2+^ supplements;	Acidosis improvement
20	Paliwal et al.–2018 [31]	F	20	Sjögren syndrome + SLE	Hypokalemic paralysis, nephrocalcinosis, SICCA, proteinuria (1.44 g/24 h)	7.15	13.1	1.8	NS	7	-	K^+^, HCO_3_^−^ supplements;	Acidosis improvement
21	Paliwal et al.–2018 [31]	F	20	Sjögren syndrome	Hypokalemic paralysis (tetraparesis), fractures, SICCA, proteinuria (0.641 g/24 h)	7.22	11	2.9	NS	8	-	K^+^, HCO_3_^−^ and Ca^2+^ supplements;	Muscular weakness improvement
22	Schilcher et al.–2017 [32]	F	51	Sjögren syndrome	Hypokalemic paralysis, lithiasis with hydronephrosis, urinary infection, SICCA, proteinuria (0.48 g/24 h)	7.21	10.4	1.6	NS; 46	8	-	Initial treatment: K^+^ supplements, prednisone 25 mg/day, AZA 2.5 mg/kg/day, HCQ Outcome: no amelioration of dRTA Treatment after 9 months: Replacement of AZA with RTX (1 g–day 0 and day 14; followed by courses administered every 6 months) Outcome: eGFR improvement to 71 mL/min/1.73 m^2^ and dRTA correction, with no need for K^+^ supplements anymore	
23	Shahbaz et al.–2018 [20]	F	28	Sjögren syndrome	Hypokalemic paralysis, respiratory failure	7.04	12	1.5	NS	6.5	-	K^+^ supplements, CS	Muscular weakness improvement and K^+^ correction
24	Tian Du et al.–2020 [33]	F	37	Sjögren syndrome	Hypokalemic paralysis, Fanconi syndrome, proteinuria 0.69 g/24 h	7.36	20.7	1.7	1.3; 53	7.5	TIN	Initial treatment: K^+^ supplements, prednisone 30 mg/day, MMF 1.5 g/day Outcome: no amelioration of creatinine Treatment change: increasing prednisone dose to 50 mg/day, adding CYC 100 mg/day Outcome: amelioration of creatinine (0.9 mg/dl), K^+^ correction	
25	Vasquez-Rios et al.–2019 [18]	F	57	Sjögren syndrome	Hypokalemic paralysis, SICCA	7.29	16	2.5	1.3; 46	7	TIN	K^+^, HCO_3_^−^ supplements; Amiloride, HCQ, AZA 50–100 mg, Prednisone	Persistent hypokalemia and acidosis, despite receiving supplements, with severe relapse when supplements are withdrawn
26	Wang et al.–2020 [34]	F	38	Sjögren syndrome	Hypokalemic paralysis, hypocalcemia, fractures, purpura	7.33	21.7	2.8	0.62; 115	7.5	-	K^+^ and Ca^2+^ supplements; Prednisone 1 mg/kg/day, CYC 0.5 g/m^2^ IV monthly	NS
27	Wang et al.–2020 [34]	F	36	Sjögren syndrome	Hypokalemic paralysis (since 16 years old), SICCA, fractures, purpura, anemia, proteinuria	7.19	15.2	2.9	2.56; 23	7	-	NS	NS
28	Wang et al.–2020 [34]	F	25	Sjögren syndrome	Hypokalemic paralysis (since 16 years old), SICCA, anemia, thrombocytopenia, hypocalcemia	7.03	9.9	2.5	1.57;	6.5	-	NS	NS
29	Wang et al.–2020 [34]	F	28	Sjögren syndrome	Fatigue, SICCA	7.13	15.6	2.9	1.14; 66	7.5	-	NS	NS
30	Yuvaraj et al.–2018 [26]	F	33	Sjögren syndrome	Hypokalemic paralysis (onset after first pregnancy), PREGNANT, SICCA, lithiasis, hypocalcemia, nephrocalcinosis	NS	14	2.4	0.55; 124	8	-	K^+^, HCO_3_^−^ supplements	Preterm labor (36 weeks), fetal bradyarrhythmia, fetal growth retardation
31	Zhou et al.–2019 [16]	F	29	Sjögren syndrome	Hypokalemic paralysis, SICCA, anemia, STILLBIRTH, lithiasis	7.28	10.6	2.1	1.34; 54	6	-	K^+^ supplements	NS
32	Elitok et al.–2020 [35]	F	60	Primary biliary cirrhosis	Progressive chronic kidney disease, leukocyturia	7.23	11.1	3.3	1.93; 28	7	TIN	NS	NS
33	Dong et al.–2018 [36]	F	32	Primary biliary cirrhosis	Biliary lithiasis, nephrocalcinosis, fatigue	7.34	17.7	2.42	0.79; 99	7.5	-	K^+^ supplements	NS
34	Duarte Silveira et al.–2022 [37]	F	29	Autoimmune hepatitis	Renal lithiasis	7.30	17	3.4	1.03; 74	7	-	K^+^ supplements	NS
35	Duarte Silveira et al.–2022 [37]	F	67	Rheumatoid arthristis	Peripheral arterial disease, hypomagnesemia	7.38	15.8	3.1	0.51; 100	6.5	-	K^+^ supplements	NS
36	Duarte Silveira et al.–2022 [37]	F	30	Sjögren syndrome	Recurrent arthritis	7.36	17	3.3	0.70; 117	7	-	K^+^ supplements	NS
37	Bruns et al.–2020 [38]	F	17	Sjögren syndrome	Recurrent muscular weakness, Hashimoto’s thyroiditis, celiac disease, central pontine myelinolysis, proteinuria	NS	13.3	1.8	NS	7	TIN	K^+^ supplements; Prednisone 60 mg/day, tapered in 2 months	Remission of proteinuria; long term need for supplements

TIN, tubulointerstitial nephritis; Nr, current number of patients; NS, not specified; HCQ, hydroxychloroquine; CS, corticosteroids; RTX, rituximab; CYC, cyclophosphamide; MMF, mycophenolate mofetil; AZA, azathioprine; eGFR, estimated glomerular filtration rate; SLE, systemic lupus erythematosus; SICCA–SICCA syndrome (referring to exocrine glands involvement).

**Table 2 biomedicines-10-02131-t002:** Treatment and outcome of patients with Sjögren’s syndrome, that is associated with TIN ± dRTA.

A. Treatment and Outcome of Patients with TIN Associated with Sjögren’s Syndrome				
Author	No. of Patients	Presentation (N)	Immunosuppressive Treatment (N; %)	Outcome (N; % of Treatment Subcategory)
Maripuri et al.–2009 [1]	19	PRF (1) AKI (4) CKD (12) ESRD (2)	Corticosteroids (9–47.3%)	Stable renal function (3; 33.3%) ↑ eGFR with > 25% (4; 44.5%) ESRD (1; 11.1%) NS (1; 11.1%)
			Corticosteroids + other immunosuppressive therapy (9–47.3%): HCQ/HCQ, RTX/RTX/PE/CYC/CYC, MMF	Stable renal function (3; 33.3%) ↑ eGFR with > 25% (3; 33.3%) ↓ eGFR with > 25% (2; 22.3%) ESRD (1; 11.1%)
			None (1)	Stable renal function
Kidder et al.–2015 [2]	15	PRF (1) AKI (8) CKD (6)	Corticosteroids (7–46.6%)	Stable renal function (2; 28.57%) ↑ eGFR with > 25% (2; 28.57%) ESRD (3; 48.86%)
			Corticosteroids + other immunosuppressive therapy (3–20%): AZA/MMF/HCQ	Stable renal function (1; 33.34%) ↑ eGFR with > 25% (2; 66.66%)
			None (5–33.34%)	↑ eGFR with > 25% (3; 60%) ↓ eGFR with > 25% (1; 20%) ESRD (1; 20%)
Goules et al.–2019 [51]	8	PRF (1) CKD (6)	Corticosteroids + other immunosuppressive therapy (6–75%): +MMF/AZA, MMF, RTX/HCQ, MMF, MTX/MTX, infliximab, adalimumab, certolizumab	Stable renal function (3; 37.5%) ↑ eGFR with > 25% (2; 25%) ↓ eGFR with > 25% (3; 37.5%)
			MMF (2–25%)	
Jasiek et al.–2016 [53]	64	NS	Corticosteroids	↑ eGFR from 39.8 mL/min/1.73 m^2^ to 49.3 mL/min/1.73 m^2^ after 6 months of follow-up (*p* < 0.001) *, **: no significant difference compared to corticosteroids administered alone regarding the outcome (↑ eGFR with at least 20%); p = 0.9, respectively p = 0.5
			Corticosteroids + other immunosuppressive therapy (*)	
			RTX (**)	
Shen et al.–2017 [54]	70	NS	Corticosteroids (56–80%)	↑ eGFR with 21.35 ± 19.63 mL/min/1.73 m^2^ vs. 2.72 ± 19.11 mL/min/1.73 m^2^ after 12 months of follow-up (*p* < 0.001) in the group receiving CYC
			Corticosteroids + CYC (14–20%)	
**B. Treatment and outcome of patients with DRTA associated with Sjögren’s syndrome and TIN**				
Maripuri et al.–2009 [1]	7	CKD (2), AKI (1), NF (1)	Corticosteroids (4; 57.14%)	Stable renal function (1; 25%), ↑ eGFR with > 25% (2; 50%) NS (1; 25%)
		CKD (1)	Corticosteroids + RTX (1; 14.28%)	↑ eGFR with > 25%
		CKD (1)	Corticosteroids + HCQ (1; 14.28%)	↓ eGFR with > 25%
		CKD (1)	PE (1; 14.28%)	Stable renal function
Evans et al.–2016 [52]	8	NS	MMF + HCQ (1; 12.5%)	↑ eGFR
			Corticosteroids + MMF (3; 37.5%)	Stable renal function (1; 33.33%), ↑ eGFR (2; 66.67%)
			Corticosteroids + MMF + HCQ (3; 37.5%)	Stable renal function (1; 33.33%), ↑ eGFR (1; 33.33%), ↓ eGFR (1; 33.33%)
			AZA (1; 12.5%)	Stable renal function
Kidder et al.–2015 [2]	2	AKI (1)	Corticosteroids	ESRD
		CKD (1)	None	↓ eGFR with > 25%
Goules et al.–2019 [51]	1	PRF	Corticosteroids + MMF + HCQ + MTX	↓ eGFR with > 25%

TIN, tubulointerstitial nephritis; N, number of patients; PRF, preserved renal function; AKI, acute kidney injury; CKD chronic kidney disease; ESRD, end stage renal disease; NS, not specified; HCQ, hydroxychloroquine; RTX, rituximab; PE, plasma exchange; CYC, cyclophosphamide; MTX, methotrexate; MMF, mycophenolate mofetil; AZA, azathioprine; eGFR, estimated glomerular filtration rate. ↑ increasing; ↓ decreasing.

## Data Availability

Not applicable.

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
