# Peer review of "Distal Renal Tubular Acidosis in Patients with Autoimmune Diseases—An Update on Pathogenesis, Clinical Presentation and Therapeutic Strategies"

_biomedicines, 2022, doi:10.3390/biomedicines10092131_

Round 1

Reviewer 1 Report

This is an interesting Review.

Minor suggestions

I recommend clarifying in the Introduction (page 1), lines 5-6, that for establishing the diagnosis of incomplete distal tubular acidosis, the gold standard urinary acidification test is the ammonium chloride test, and that furosemide and fludrocortisone (FF) test is an alternative test well tolerated but less accurate.

The authors specify this subject on page 2, lines 9-10, but they only say that the FF is a test well tolerated. It should be added that this test is well tolerated but less sensitive and accurate than the ammonium test, as showed by Both T et al. in 2015 (Rheumatology 2015; 54: 933-39).

Author Response

Thank you very much for the suggestions. We have commented on the difference between Furosemide/Fludrocortisone test and ammonium chloride test regarding tolerance and sensitivity.  All the modifications are marked in the manuscript with the “track changes” option to be easily identified. You will find the modified version of the manuscript in the attachment.   

Reviewer 2 Report

In recent years, knowledge of the molecular mechanisms behind acid secretion has improved, helping the diagnosis of dRTA. The primary or inherited form of dRTA is mostly diagnosed in infancy, childhood, or young adulthood, while the acquired secondary form, as a consequence of other disorders or medications, can happen at any age, although it is more commonly seen in adults. It is now known that dRTA can have several, highly variable long-term consequences. The authors presented an interesting review of individual case reports of dRTA in patients with Sjogren's syndrome and SLE published in the last 5 years.

The title of this paper refers to autoimmune diseases, however the authors limited the description to cases of Sjogren's syndrome and SLE. Therefore, it would be worth expanding the description and discussion with case reports with other autoimmune diseases, e.g. rheumatoid arthritis, autoimmune hepatitis, primary biliary cirrhosis, thyroiditis or scleroderma. The pathogenesis and treatment of these diseases vary and may affect the management of dRTA.

If, on the other hand, the authors want to limit themselves only to Sjogren's syndrome and SLE - they should include it in the title and not make generalizations - which is associated with corrections also in the text.

Author Response

Thank you very much for the suggestions. We have expanded our description of case reports and discussion with two cases of primary biliary cirrhosis, one case of autoimmune hepatitis, one case of rheumatoid arthritis and one case of Hashimoto thyroiditis in association with Sjögren’s syndrome. All the cases have been published in the last five years, as our initial selection of cases. We presented also data regarding possible mechanisms of pathogenesis derived from one of the cases with primary biliary cirrhosis. All the modifications are marked in the manuscript with the “track changes” option to be easily identified. You will find the modified version of the manuscript in the attachment.   

Reviewer 3 Report

The manuscript is interesting and well written. However, the authors do not discuss the role of of immune system in the pathogenesis of distal renal tubular acidosis. I suggest to discuss the role of Th17 cells, IL-31/IL-33 axis and soluble HLA class I (see and add as references papers by Murdaca etL concerning Th17 in chronic inflammatory immune mediated diseases, concerning IL-31/IL-33 axis and HLA-G in HIV)

Author Response

Thank you very much for the suggestions. We have included in the manuscript data regarding the role of Th 17 cells and the signature molecule IL-17 and IL-31/IL-33 axis in autoimmunity, stating the importance of further research in the field in order to establish if there is a role of these molecules in the development and evolution of distal renal tubular acidosis and also if molecules like Secukinumab might be useful for these patients. All the modifications are marked in the manuscript with the “track changes” option to be easily identified. You will find the modified version of the manuscript in the attachment.   
